# Silicosis, tuberculosis and silica exposure among artisanal and small-scale miners: A systematic review and modelling paper

**Patrick Howlett**[1]*, **Hader Mousa**[2], **Bibie Said**[3], **Alexander Mbuya**[3], **Onn Min Kon**[1], **Stellah Mpagama**[3‡], **Johanna Feary**[1‡]

1 National Heart & Lung Institute, Imperial College London, London, United Kingdom, 2 Centre for Occupational and Environmental Health, Kigali, Rwanda, 3 Kibong'oto Infectious Disease Hospital, Kilimanjaro, Tanzania

‡ SM and JF are joint final authors on this work.
* patrick.howlett@gmail.com

## Abstract

An estimated 44 million artisanal and small-scale miners (ASM), largely based in developing economies, face significant occupational risks for respiratory diseases which have not been reviewed. We therefore aimed to review studies that describe silicosis and tuberculosis prevalence and respirable crystalline silica (RCS) exposures among ASM and use background evidence to better understand the relationship between exposures and disease outcomes. We searched PubMed, Web of Science, Scopus and Embase for studies published before the 24th March 2023. Our primary outcome of interest was silicosis or tuberculosis among ASM. Secondary outcomes included measurements of respirable dust or silica, spirometry and prevalence of respiratory symptoms. A systematic review and narrative synthesis was performed and risk of bias assessed using the Joanna Briggs Prevalence Critical Appraisal Tool. Logistic and Poisson regression models with predefined parameters were used to estimate silicosis prevalence and tuberculosis incidence at different distributions of cumulative silica exposure. We identified 18 eligible studies that included 29,562 miners from 13 distinct populations in 10 countries. Silicosis prevalence ranged from 11 to 37%, despite four of five studies reporting an average median duration of mining of <6 years. Tuberculosis prevalence was high; microbiologically confirmed disease ranged from 1.8 to 6.1% and clinical disease 3.0 to 17%. Average RCS intensity was very high (range 0.19–89.5 mg/m$^3$) and respiratory symptoms were common. Our modelling demonstrated decreases in cumulative RCS are associated with reductions in silicosis and tuberculosis, with greater reductions at higher mean exposures. Despite potential selection and measurement bias, prevalence of silicosis and tuberculosis were high in the studies identified in this review. Our modelling demonstrated the greatest respiratory health benefits of reducing RCS are in those with highest exposures. ASM face a high occupational respiratory disease burden which can be reduced by low-cost and effective reductions in RCS.

**Data Availability Statement:** All data is available within the article and is publically available via articles included.

**Funding:** PH is supported by an MRC Clinical Research Training Fellowship (MR/W024861/1). SM is supported by the EDCTP2 program, European Union (grant number TMA 2016SF-1463-REMODELTZ). The funders had no role in study design, data collection and analysis, decision to publish, or preparation of the manuscript.

**Competing interests:** The authors have declared that no competing interests exist.

## Introduction

Silicosis is an incurable fibrotic lung disease caused–in a dose-dependent manner–by exposure to respirable crystalline silica (RCS). It usually presents decades after exposure, but high exposures can lead to accelerated disease and respiratory failure within 1–2 years [1]. It represents an under-estimated emerging public health threat in developing economies [2]. In 2021, 1.6 million people died from tuberculosis (TB), the majority of whom lived in developing economies [3]. Supported by mechanistic evidence [4], the risk of TB in those with silicosis is 2.9–5.6 times higher than those without silicosis and RCS exposure is a further, independent risk factor for TB [5]. TB in those with silicosis is associated with poor outcomes [6, 7]. Human immunodeficiency virus (HIV) and RCS are risk factors for TB that combine multiplicatively with silicosis [8, 9]. The structure of these associations is represented in a directed acyclic graph (Fig 1). Silica exposure is also associated with lung cancer, chronic obstructive pulmonary disease, respiratory symptoms, autoimmune disease and renal disease [10]. In view of these risks, many countries have enacted strict RCS exposure limits. For example, the United States Occupational Safety and Health Administration enforces a permissible exposure limit of 0.05 mg/m$^3$ [10–12].

An estimated 44 million artisanal and small-scale miners (ASM) worldwide are almost exclusively based in developing economies [13]. Despite the term's widespread usage no standard definition of ASM exists; a practical approach may involve one or more of a group of shared features [14]. The sector's key role in supporting development is recognised by the United Nation's sustainable development goals, the World Bank and African Union [15]. Small-scale mining represents a large and integral component of the world's economy, for example producing 20% of global gold [13, 16].

Despite the understanding that health risks are high and occupational health services sparse [14, 17], the true occupational risks of ASM are not well described. A recent scoping review found relatively few studies, with a focus on mercury exposure in gold mining. In particular, dust exposure and respiratory and communicable diseases were under-researched [16]. Many studies of large-scale miners (LSM) have been performed, however, in this systematic review we focus only on ASM populations as we believe the exposures and occupational respiratory outcomes are distinct from LSM. More precisely, we hypothesise that, among ASM, high RCS exposures lead to a high prevalence of silicosis and tuberculosis which is associated with abnormal spirometry and a high respiratory symptom burden. Higher background TB and HIV rates compound these exposure risks. We are not aware of a previous systematic review that investigates this hypothesis.

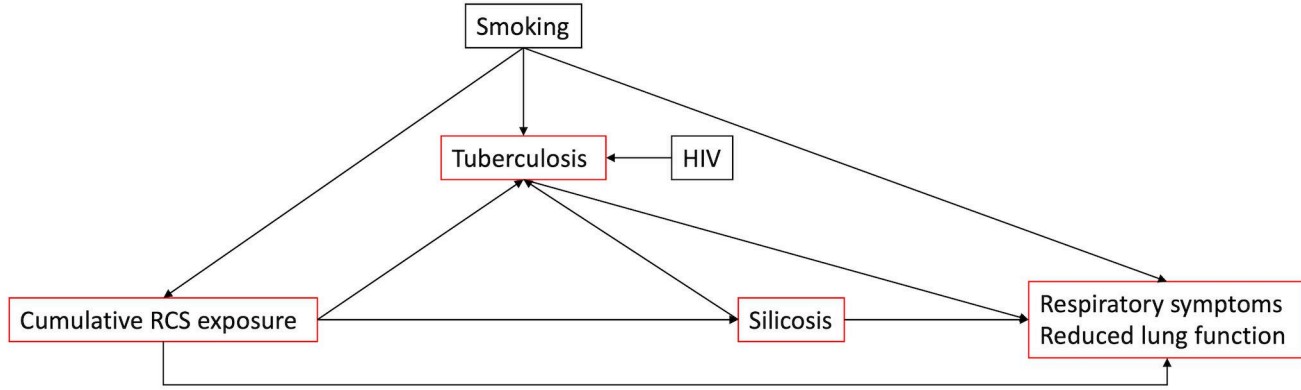

**Fig 1. A simplified directed acyclic graph (DAG) describing the relationship between silica exposure, silicosis, tuberculosis, HIV, respiratory symptoms and lung function.** The boxes in red are the outcomes included in this systematic review.

Observational studies linking individual exposures, silicosis risk and TB incidence among LSM are comprehensive and generally good quality [18, 19], although none adjust for both RCS and HIV as implied by our conceptual framework (Fig 1). We are not aware of similar observational data for ASM, in whom silica exposure may be higher, and no interventional studies linking dust reduction to respiratory outcomes in any setting have been reported. In the absence of such data, extrapolating from LSM studies and using our conceptual framework (Fig 1) may improve our understanding of the determinants of silicosis and TB.

In this systematic review therefore, among ASM, we aim to describe 1) the prevalence of silicosis and tuberculosis, 2) occupational exposures to respirable dust and silica 3) spirometry findings, and 4) the prevalence of respiratory symptoms. As we expect few eligible studies, we believe the holistic approach of this review improves our ability to build a picture of the respiratory health of ASM and, as the outcomes are related (Fig 1), improve our ability to infer plausibility of respective findings. In the second part of our study, we aim to model silicosis prevalence and TB incidence with respect to RCS exposures compatible with our review and risks observed in previous studies. This will allow observation of disease risks in counter-factual scenarios and contextualise silicosis and TB estimates from our systematic review.

## Methods

### Ethics statement

This review utilised only published data, as such no ethics clearance was required.

### Search strategy and selection criteria

We searched PubMed, Embase, Web of Science and Scopus for studies published prior to 24th March 2023. Our search strategy was broad and structured using population terms (e.g (mining or mine*).mp) and respiratory terms (e.g. exp/. Respiratory system); the full strategy is in S1 File. The review was registered at PROSPERO (https://www.crd.york.ac.uk/prospero/) ID: CRD42023394100.

We included original research papers that recruited small-scale miners and reported one or more of (1) silicosis prevalence, (2) tuberculosis prevalence, (3) a measure of respirable crystalline silica or dust exposure, (4) spirometry values, or (5) respiratory symptom prevalence. We excluded case reports or series, systematic reviews and meta-analyses and qualitative studies. We excluded papers that focused on processing of materials (e.g. stone carvers).

We used the Covidence platform to manage the study inclusion process. After merging the four database searches and removing duplicates, two reviewers (PH and either HM or BS) independently reviewed the titles and abstracts and subsequently full texts in parallel, with any disagreements resolved by group discussion. Our PRISMA checklists are provided in S2 File.

### Data analysis

One team member (either PH or HM) independently extracted study characteristics and performed quality assessment; this was checked by another team member and disagreements resolved by group consensus. Our data extraction and quality assessment tools are presented in the S3 File. We used the categorical cut off reported in the study to define silicosis prevalence and extracted self-reported silicosis separately. Microbiological TB prevalence required any of smear microscopy, GeneXpert or culture positivity. Clinical prevalence additionally included treatment based on symptoms or imaging. Concentration (mg/m$^3$) and duration of measurement of respirable silica dust exposures were extracted. Spirometry values were extracted as presented. We chose *a priori* to extract prevalence of shortness of breath (at rest),

cough and wheeze as respiratory symptoms. In cases where additional information or clarification was needed authors were contacted.

For quality assessment, we used the Joanna Briggs Institute Prevalence Critical Appraisal Tool [20] (see S3 File). We changed from the adapted Newcastle-Ottawa scale, described in our protocol, as we found it more applicable, rigorous and relevant for prevalence studies [21]. We tested for bias separately for each outcome.

### Data synthesis

Using the meta package in R (version 4.2.2) we performed a meta-analysis of three outcomes; silicosis prevalence, clinical TB prevalence and microbiological TB prevalence. We used a logit transformation of prevalence and then a generalised linear mixed-effects model to estimate the pooled effects [22]. Random-effects allowed for expected between study heterogeneity. The variance of the true effect was estimated using maximum likelihood method. Individual study confidence intervals were estimated with the Clopper-Pearson method. As the number of studies was low and a priori heterogeneity expected to be high, the Knapp-Hartung adjustment was applied [23]. Heterogeneity was assessed using the $I^2$ statistic. Prevalence estimates were visualised using forest plots and systematic differences according to variance by funnel plots.

### Modelling of silicosis prevalence and annual TB incidence

In the logistic regression Eq (1) $p_1$ (the probability of silicosis) is calculated using the following parameters: Baseline silicosis risk when RCS exposure is zero was estimated at 2% [19, 24]. $B_1$ is the odds of silicosis per 1 mg/m$^3$-years increase in cumulative exposure of RCS and equal to 1.3 [25–28]. The probability of silicosis according to cumulative RCS values, at different odds ratios is visualised in S1 Fig. The distribution of cumulative RCS exposure (mg/m$^3$-years) has a positive skew, which diminished as the mean concentration increased. Illustrative distributions are provided in S2 Fig.

Eq 1:

$$\text{Logit}(p_1) = \text{Baseline silicosis risk} + (B_1 * \text{Cumulative RCS exposure}) \tag{1}$$

In the Poisson regression Eq (2), the following parameters are used to calculate $p_2$ (the probability of tuberculosis): Baseline annual tuberculosis incidence was 200 cases per 100,000 persons [29]. $B_1$ is the odds of TB per 1 mg/m$^3$-years increase in cumulative RCS exposure and was set at 1.05 [5, 9]. $B_2$ is the odds of TB in the presence of silicosis compared to those without silicosis and was set at 4 [5]. Silicosis prevalence was determined by Formula 1. $B_3$ is the odds of TB in the presence of HIV infection compared to those without HIV infection and was set at 4 [8, 30]. HIV prevalence is randomly distributed in the population with a default prevalence of 2%.

Eq 2:

$$\text{Log}(p_2) = \text{Baseline annual TB incidence} + (B_1 * \text{RCS exposure}) + (B_2 * \text{silicosis}) + (B_3 * \text{HIV}) \tag{2}$$

Detailed justification for the parameter assumptions made in both models are described in the S4 File.

To allow interaction, this model is published using the shiny package and is available (https://phowlett.shinyapps.io/sil_tb_app/). All code used in this study is publicly available (https://github.com/pjhowlett/asm_sr/tree/main).

## Results

### Study selection

From a total of 357 titles and abstracts, 56 studies were full-text screened and 18 eligible studies of 13 distinct populations from 10 countries included in our analysis (Fig 2) [31–48]. Of these studies, six reported on silicosis prevalence [31–36], six on tuberculosis prevalence [34–39], eight on dust or respirable silica exposures [31, 33, 40–45], three on spirometry findings [40, 47, 48] and three on respiratory symptoms [33, 44, 46]. There was moderate agreement for

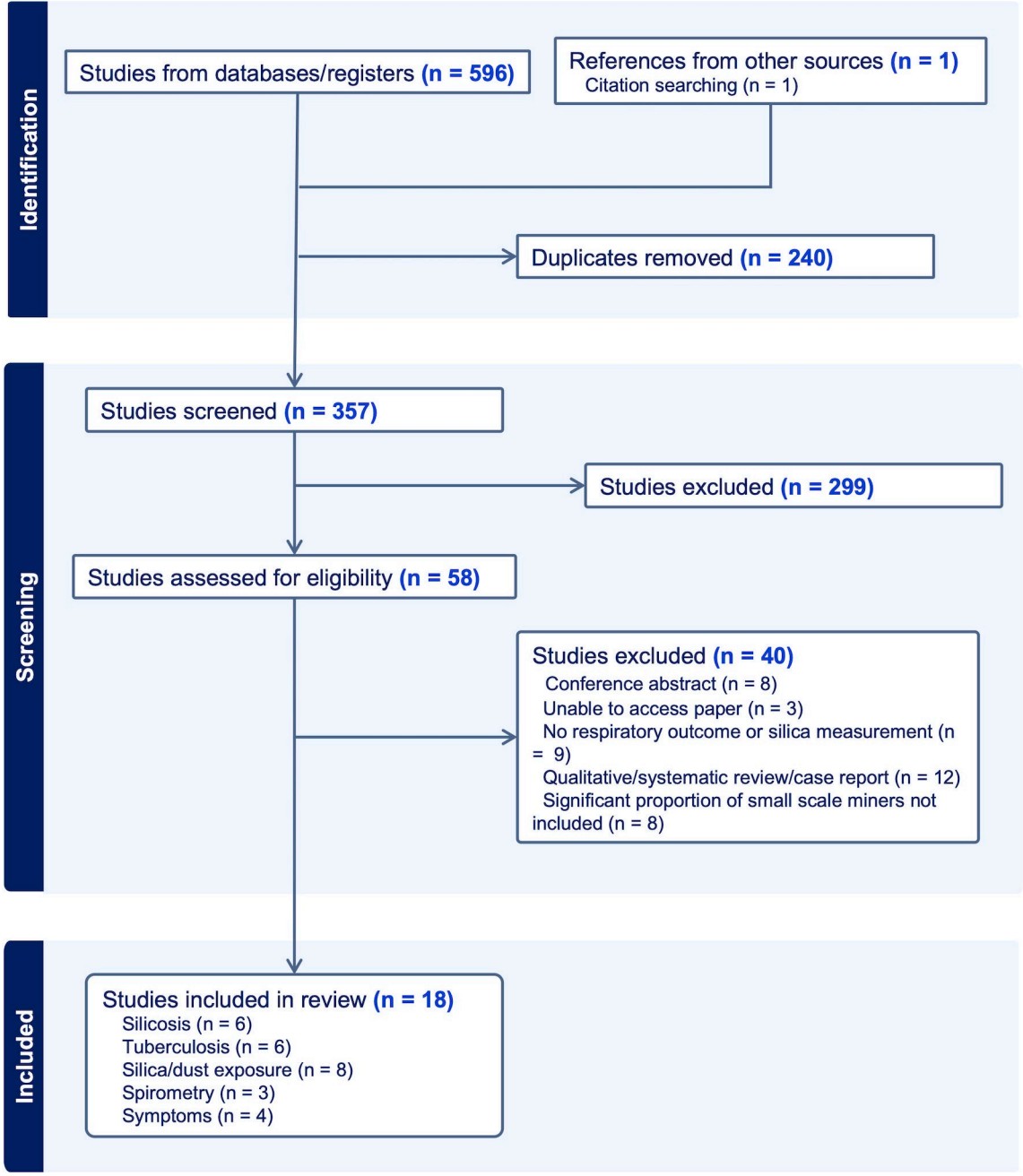

**Fig 2. PRISMA diagram of the identification, screening and inclusion process for our systematic review.**

title and abstract review, Cohen's kappa of 0.56 (HM) and 0.49 (BS). For full text reviews, Cohen's kappa was 0.80 (HM) and 0.28 (BS), indicating at least fair agreement.

## Study characteristics

The number of ASM enrolled in all 14 clinical studies ranged from 18 to 21,136 participants (median 441, IQR 258–867) with an overall total of 29,562 participants. There were 14 studies from Africa [34–47], two studies from Asia [31, 48] and two from South America [32, 33]. Most of the included studies were in gold (10/18) and gemstone (5/18) mining communities and were exclusively underground (10/18) or both under and above ground ASM (5/18). In those reporting age, the summary statistic ranged between 24.4 and 40.1 years. In studies reporting sex, five studies included only male miners [31–33, 36, 40] while in five other studies the majority (59–85%) were male [34, 35, 37, 46, 48]. All studies except one were of active miners. The mean or median duration of exposure ranged between 4.0–7.3 years in all bar one study [32]. The prevalence of smoking ranged from 16–58% in eight studies [31–36, 40, 44, 46, 48]. Given the importance of this finding, it is summarised in S1 Table.

## Silicosis prevalence, RCS exposures and latency

The prevalence of radiologically defined silicosis was reported in five studies [31, 32, 34–36] and ranged between 11–37%, with a total of 1116/5572 cases (see Table 1 and Fig 3). The pooled estimate was 23.9% (95% 13.9, 38.0) however the heterogeneity was very high ($I^2$ 97%). Four studies defined silicosis as ILO $\geq$1/0 with a mixture of trained and un-trained readers [31, 32, 34, 35]; one study used an experience radiologist diagnosis [36]. One study reported found a self-reported silicosis prevalence of 27/258 (10.5%) [33].

One study of gold miners in China [31] found a silicosis prevalence of 170/583 (29.1%) with a median RCS exposure of 89.5 mg/m$^3$ (range: 70.2–108.8 mg/m$^3$). A study of Amethyst miners in Brazil reported a median RCS exposure of 1.6 mg/m$^3$ [33]; self-reported silicosis prevalence was 27/258 (10.5%) although previous radiological prevalence in the same population was 129/348 (37%) [32]. Median RCS exposures in gemstone miners in Tanzania were 1.0 mg/m$^3$ (IQR 1.2–3.2) and 2.4 mg/m$^3$ (IQR 0.7–1.4) dependent on the activity [41] and silicosis prevalence in a random sample from the population was 99/330 (30%) [36].

Four of the six studies reported a mean or median duration were below 5.6 years [31, 34–36]. One study found all (32/348, 12.1%) participants attending annual screening who were temporarily absent or retired had silicosis [32].

## Tuberculosis

The prevalence of microbiologically defined TB among ASM was reported in six studies [34–39], all from the African continent, and ranged from 1.8–6.1% with a total of 618/27,171 cases of TB (see Table 1 and Fig 4). Clinically defined TB was reported in 4 of these studies [34, 35, 37, 39] and ranged from 3.0–17% prevalence with a total of 1039/25,997 cases (see Fig 4). The pooled estimates for microbiological and clinical TB were 3.5% (95% CI 1.7, 7.1%) and 6.2% (95% CI 2.0, 17.6%), respectively, with very high heterogeneity of both estimates ($I^2$ 98% and 99%, respectively). For microbiological confirmation of TB, studies used GeneXpert NAAT testing, although one study reported using smear microscopy if Xpert was not available [39]. Two studies included miners attending a facility and outreach screening [34, 35], three recruited from outreach-based screening only [37–39] and one used randomised sampling [36]. Notably, one study found miners attending a facility were reported to have an increased risk of TB and silicosis, prevalence ratio 10.3 (95% CI 7.6,14.0) and 1.5 (95% CI 1.3,1.7) respectively, compared to outreach [35]. Only one study reported that study just under half (49.6%)

**Table 1. Characteristics of studies reporting silicosis and/or tuberculosis prevalence estimates among ASM.**

| Author, year | Study country, period | Study design | Population | Sample size | Sampling method | Silicosis definition | TB screening and diagnosis methods | Age (years), sex (%) | Duration of mine exposure (years) | Silicosis prevalence (%) | TB prevalence (%) | Notes |
|---|---|---|---|---|---|---|---|---|---|---|---|---|
| **Silicosis only** | | | | | | | | | | | | |
| Tse, 2007 [31] | China, 1997–2001 | Cross-sectional | Retired underground gold ASM rock-drillers | 583 | Complete sample of rock-drillers | Consensus of two readers; ILO >1/0 | - | All male. Age at first service 24.4 SD +/- 6.7 years | 5.6 (SD 2.4) years | 170/583 (29.1%) | - | Follow up 3–7 years after finishing work |
| Souza, 2017 [32] | Brazil, 2013–2014 | Cross-sectional | Current and retired underground precious stone ASM | 348 | Sequential sample of registered miners attending annual screening | Two readers (kappa = 0.91); ILO >1/0 | - | Mean 40.1 (SD +/- 11.9) years, all male | 20.4 (SD 12.8) years | 129/348 (37%) | - | All absent (17/348, 4.9%) and retired (25/348, 7.2%) workers had silicosis |
| Souza, 2021 [33] | Brazil, 2017–2018 | Cross-sectional | Current underground precious stone ASM | 258 | Miners from 49/277 randomly selected mines | Self-reported silicosis | - | Mean 40 (SD +/- 16.0) years, all male | 177/258 (69%) ≤20 years of work | 27/258 (10.5%) *self-reported | - | |
| **Silicosis and TB** | | | | | | | | | | | | |
| Moyo, 2021 [34] | Zimbabwe, 2020–2021 | Cross-sectional | Current above and below ground gold ASM | 514 | Miners attending TB outreach screening and hospital occupational health clinic | Medical officer interpretation, checked by specialist physician; ILO >1/0 | Symptom and CXR screen. One Xpert sample | Mean 37.0 (SD +/- 12.7) years, 435/514 (85%) male | 4 (IQR 0.02–33) years. 352/514 (69%) < 10 years | 52/464 (11.2%) | Microbiological: 9/422 (2.1%) Clinical: 17/422 (4.0%) | HIV prevalence 90/373 (24%) |
| Moyo, 2022 [35] | Zimbabwe, 2020–2022 | Cross-sectional | Above or underground gold and chrome ASM | 3950 | Miners attending TB outreach screening and hospital occupational health clinic | Medical officer interpretation, checked by specialist physician; ILO >1/0 | Symptom and CXR screen. One Xpert Ultra sample | Mean 35.5 (SD +/- 12.1) years, 3245/3950 (85%) male | 5 (IQR 2.8-10.8) years | 666/3547 (18.8%) | Microbiological: 83/3547 (2.5%) Clinical: 240/3547 (6.8%) | HIV prevalence 460/2568 (18%) |
| Mbuya, 2023 [36] | Tanzania, 2019–2021 | Cross-sectional | Below ground gemstone ASM | 330 | 15 randomly sampled miners from 22 randomly chosen mines | As defined by radiologist with significant occupational experience | Systematic testing with one Xpert test | Median 35.0 (IQR 30.0–44.0), All male | Median 5.0 (IQR 4.0–7.0) | 99/330 (30%) | Microbiological: 20/330 (6.1%) | HIV prevalence 6/330 (1.8%). Community TB prevalence 26/330 (7.9%) |
| **TB only** | | | | | | | | | | | | |

(*Continued*)

**Table 1.** (Continued)

| Author, year | Study country, period | Study design | Population | Sample size | Sampling method | Silicosis definition | TB screening and diagnosis methods | Age (years), sex (%) | Duration of mine exposure (years) | Silicosis prevalence (%) | TB prevalence (%) | Notes |
|---|---|---|---|---|---|---|---|---|---|---|---|---|
| Rambiki, 2020 [37] * | Malawi, 2019 | Cross-sectional | Active under and above ground ASM | 934 | Miners attending TB screening programme | - | Symptom screen then CXR, two sputums initial smear; if negative Xpert | - | - | - | Microbiological: 46/892 (5.2%) Clinical: 152/892 (17%) | LSM TB prevalence 22/1032 (2.1%) |
| Ohene, 2021 [38] | Ghana, 2017–2018 | Cross-sectional | Under and overground gold ASM | 844 | Miners attending outreach screening program | - | Symptom screen and CXR. One Xpert sample | - | - | - | Microbiological: 23/844 (2.7%) - | Community TB prevalence 72/9597 (0.7%) |
| Abeid, 2022 [39] | Tanzania, 2017–2020 | Cross-sectional | Underground gold ASM | 21,136 | Miners attending TB screening programme | - | Symptom screen. One Xpert or smear sample | - | - | - | Microbiological: 380/21,136 (1.8%) Clinical: 630/21,136 (3.0%) | HIV prevalence 110/494 (22.7%). Community TB prevalence 904/100,461 (0.9%) |

Abbreviations: ILO = International Labour Organisation. ASM = Artisanal and small-scale miners. LSM = Large-scale miners. SD = standard deviation. IQR = interquartile range. CXR = chest X-ray. TB = Tuberculosis. HIV = Human immunodeficiency virus

* In the study of Rambiki et al. in the whole study of ASM and LSM the most frequent category was 570 participants aged 25–34 (28%), 1426/2013 (71%) were male and most worked for between 0–3.5 years (1125/2013, 56%). HIV prevalence was 29/1438 (2.0%).

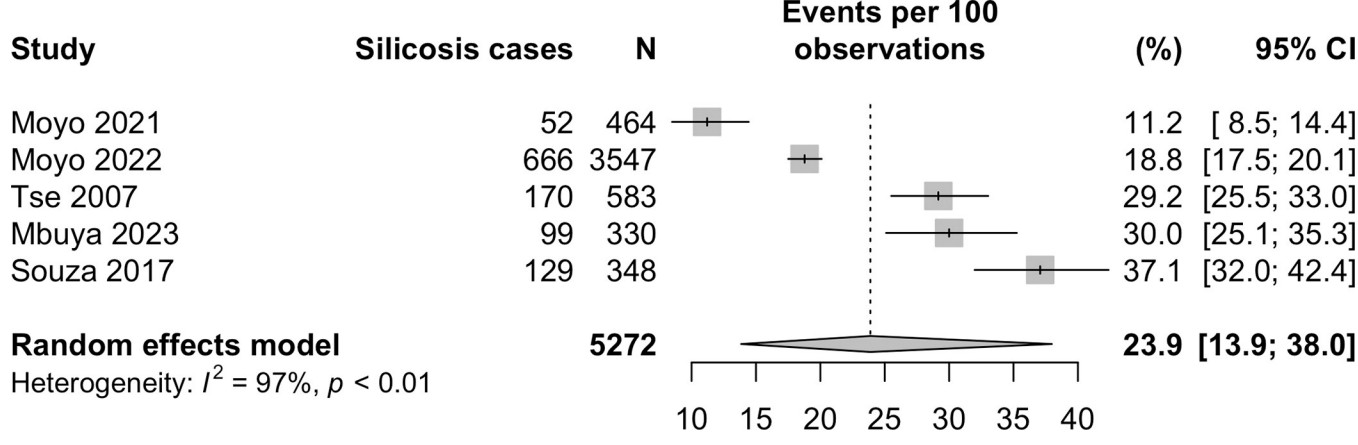

**Fig 3. Forest plot of silicosis prevalence in studies of ASM miners.**

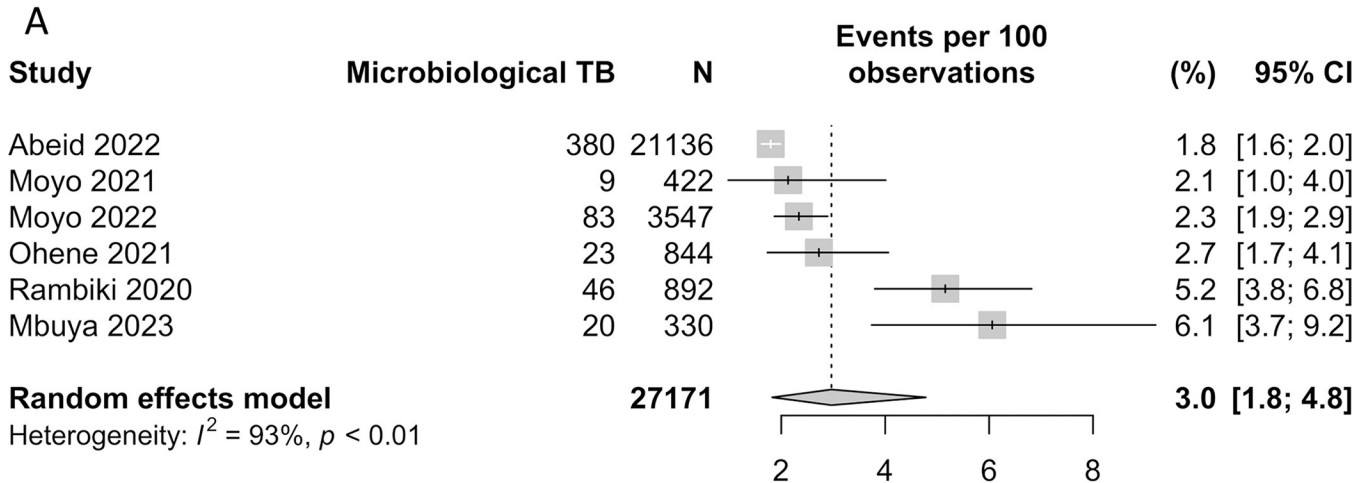

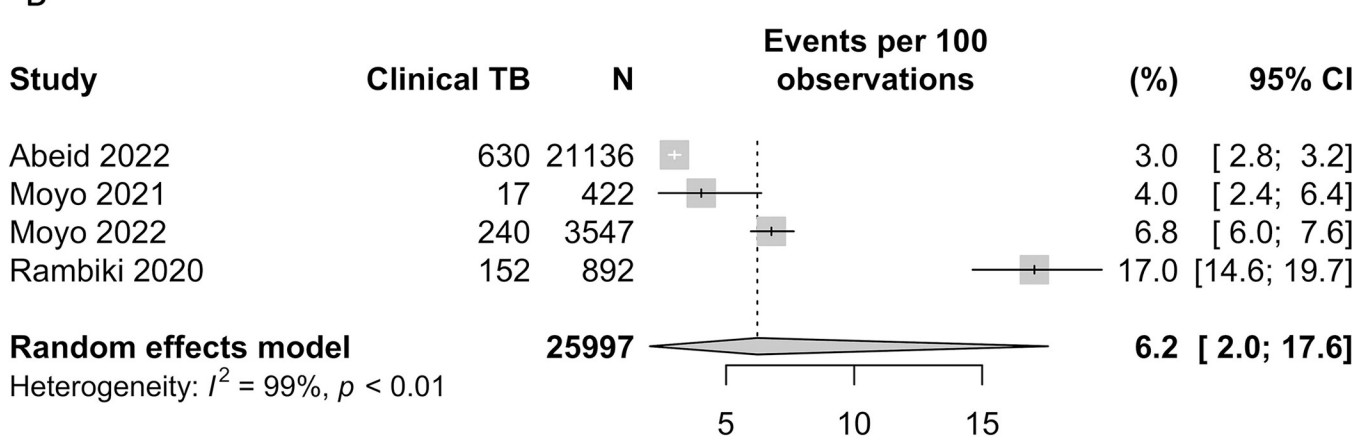

**Fig 4.** Forest plot of A. microbiological tuberculosis prevalence and B. clinical tuberculosis in studies of ASM miners.

had sputum samples available when clinically indicated [39]; other studies were not able to provide these results when requested. HIV prevalence in the ASM populations varied significantly between 1.8–24% though was often missing (0–97% of participants) [34–37, 39]. In three studies TB prevalence was higher in ASM than in the community or LSM, however one study reported a slightly higher prevalence among community members (7.9% vs 6.1%) [36].

## Exposures

Average RCS concentrations ranged between 0.19 and 89.5 mg/m$^3$ and were reported in four different ASM communities across three countries; Tanzania [41, 42], China [31] and Nigeria [45] (see Table 2). Reported sampling times ranged between 4 to 8 hours. In one interventional study among gold miners, introducing wet methods during crushing led to an 80% reduction in respirable silica concentration from 0.81 (SD +/- SD 0.35) mg/m$^3$ to 0.06 mg/m$^3$ (SD +/- SD 0.05) [45].

## Spirometry

Spirometry results were reported by three studies, which all provided control groups [40, 47, 48] (see S2 Table). One study, from Zimbabwe, found significantly lower raw FEV1 and FVC results in ASM than both LSM and community controls and higher dust concentrations in ASM (6.0 +/- 0.5 mg/m$^3$) compared to LSM (0.5 +/- 0.1 mg/m$^3$) [40]. Two other studies demonstrated no significant difference between ASM and community controls [47, 48].

## Symptoms

Three studies reported the prevalence of shortness of breath was between 3.6–27.6% and cough between 25.2%-47.7% [33, 44, 46] (see S1 Table). Wheeze prevalence was 42.7% in a single study [44]. All respiratory symptoms were more commonly reported by miners than LSM and community controls [44, 46]. Additionally, two studies reported a high prevalence of rhinitis of 41.4% [46] and 70.4% [44].

## Study quality

Selection bias due to the lack of systematic or randomised approaches and inclusion of facility-based participants was a recurrent issue (Table 3). Within individual outcomes, validity of silicosis diagnosis and the application of outcome measurement for TB (for example, the lack of reporting of sputum availability) were recurrent issues. For exposure measurements, no study provided a systematic or randomised approach to sampling or sample size calculation. For spirometry and symptom outcomes, statistical errors, sample size issues and risk of selection bias reduced quality. The funnel plots (S3 Fig) do not suggest systematic change in outcome by standard error although inference is limited by high heterogeneity and small sample sizes.

## Modelling

First, we modelled silicosis prevalence and TB incidence at distributions of RCS with mean values between 0 and 20 mg/m$^3$-years, at stepwise increments of 0.01 mg/m$^3$-years and with a sample size of 10,000 per simulation. We repeated this at RCS and silicosis associations of OR 1.2, 1.3 and 1.5 (Fig 5A and 5B, and S3 Table). For all three odds ratios the prevalence of silicosis and TB incidence increased with cumulative RCS, with a higher rate of increase at higher odds. Second, we estimated annual TB incidence at varying baseline HIV prevalence estimates (2%, 4% and 20%) and assumed either a two- or four-fold increased odds of TB disease for in the presence of HIV infection (Fig 5C and 5D, respectively). In these models the annual

**Table 2. Characteristics of studies reporting RCS and/or dust exposure estimates among ASM.**

| Author, year | Study country, period | Study design | Population | Sample size | Sampling method | Dust measurement method | RCS measurement (mg/m³)* | Dust measurement (mg/m³) |
|---|---|---|---|---|---|---|---|---|
| Bratveit, 2003 [41] | Tanzania, 2001 | Cross-sectional | Underground gemstone ASM | 15 samples (9 for silica); 10 on day 1 and 2 –drilling and blasting—(6 for silica and 5 on day 3 –loading—(3 for silica) | Opportunistically selected miners over three consecutive days from one selected mine | Shift personal respirable air samples, gravimetric analysis and analysed using NIOSH method 7500. Sampling time 5–8 hrs. | Day 1 and 2: Median 2.4 (IQR 1.2–3.2) Day 3: Median 1.0 (IQR 0.7 to 1.4) | Day 1 and 2: Median 15.5 (IQR 10.4–21.2) Day 3: Median 4.3 (IQR 3.7–7.4) |
| Osim 1999 [40] | Zimbabwe, no date described | Exposure-control | Above and below ground chrome ASM, chrome LSM and community controls | 4 ASM sites, 4 LSM sites and 4 control sites | Not described | Samples taken from middle of each site for 4 hours with respirable gravimetric dust sampler air flow of 2L per hour for 4 hours. | | ASM: 6.0 +/- SD 0.5 LSM: 0.5 +/- SD 0.3 Community: 0.3 +/- SD 0.1 |
| Tse, 2007 [31] | China, 1997–2001 | Cross-sectional | Retired underground gold ASM rock-drillers | Unclear number of samples | Not described; government collected samples | Not described. No direct silica measurement; estimated from silica and respirable fraction. | 89.5 (range 70.2–108.8) | - |
| Gottesfeld, 2015 [42] | Tanzania, 2015 | Cross-sectional | Above and below ground gold ASM doing mining and processing | 32 samples | Mines and miners in five selected ASM villages | Shift personal respirable air samples, and stationary above ground worksites analysed using NIOSH method 7500. Sampling time above ground: 190 mins (SD 73). Below ground: 311 mins (SD 137). | Above ground: 0.19 (SD 0.20) Below ground: 16.85 (SD 8.74) | Above ground: 1.00 (SD 0.75) Below ground: 93.78 (SD 67.56) |
| Mayala, 2016 [43] | Tanzania, 2016 | Cross-sectional | Underground gemstone ASM | 5 samples | Chosen on safety, availability, trust of researchers, knowledge of managers and accessibility | A"dust detector" was set at the workface during shovelling and loading for 4 hours. | - | 8.0 (SD 1.3) |
| Leon-Kabamba, 2018 [44] | Democratic Republic of Congo, 2016 | Exposure-control | Above ground coltan ASM and government office workers | Not described | Three samples taken at each mining site and each office setting | PM 2.5 measurement using air quality monitor (Light Scattering Method– Bramc BR- AIR-329). Sampling time not described. | - | ASM sites: 0.197 (range 0.180–0.210) Office sites: 0.029 (range 0.019–0.024) |
| Gottesfeld, 2019 [45] | Nigeria, 2016 | Pre and post intervention | Underground ASM gold miners in whom a lead poisoning outbreak was reported | 29 mining and processing air samples; 18 before intervention, 11 after interventions | Voluntary participation | Shift personal respirable air samples collected using NIOSH method 7500. Sampling time dry 265–376 mins, wet spray 352–430 mins, wet stream 109–380, mining 333–393 mins. | Mining: 0.2 +/- SD 0.14 Dry processing: 0.81 +/- SD 0.7 Wet spray processing: 0.16 +/- SD 0.1 Wet stream processing: 0.06 +/- SD 0.06 | - |

(*Continued*)

**Table 2.** (Continued)

| Author, year | Study country, period | Study design | Population | Sample size | Sampling method | Dust measurement method | RCS measurement (mg/m³)* | Dust measurement (mg/m³) |
|---|---|---|---|---|---|---|---|---|
| Souza, 2021 [33] | Brazil, 2017–2018 | Cross-sectional | Current underground precious stone ASM | Unclear number of samples from 2 mines | Selected mines from random sample of 49 mines | Shift respirable personal air and fixed-point samples collected using Brazil occupational hygiene method NH008. Sampling time not described. | Median 1.6 mg/m3 +/- IQR 0.41# | Median 13.1 +/- SD 0.55 |

Abbreviations: NIOSH = National Institute for Occupational Safety and Health. ASM = Artisanal and small-scale miners. LSM = Large-scale miners. PM = particulate matter. SD = standard deviation. IQR = interquartile range.

* All silica and dust measurements are arithmetic mean values unless otherwise specified as the median

# This is presented as reported; attempts were made to contact the authors to clarify this statistic however no response was received.

incidence of tuberculosis increased with HIV prevalence. Given the multiplicative relationship between RCS, silicosis and HIV [8, 9], the effect of HIV on TB incidence becomes more pronounced at higher cumulative RCS exposures. In all scenarios, if mean cumulative RCS is reduced the corresponding reductions in silicosis prevalence or TB incidence are greatest at higher mean cumulative RCS exposures.

## Discussion

Our systematic review demonstrates that artisanal and small-scale miners face a significant occupational respiratory health burden. Silicosis prevalence ranged from 11–37%, despite four of five studies reporting a mean or median duration of mining of less than 6 years. Microbiologically confirmed TB ranged between 1.8–6.1% and clinical TB between 3.0–17%. The plausibility of these findings is increased by evidence of very high RCS exposures (0.19–89.5 mg/m³) and prevalent respiratory symptoms. A further important finding is that smoking is common among ASM (prevalence range 17–58%), highlighting a vital need for smoking cessation interventions.

A previously reported review of 10 representative LSM cohorts found median RCS exposures ranged between 0.02–0.59 mg/m³, with median lifetime exposures of 0.7–11 mg/m³-years [49]. Corresponding lifetime prevalence of silicosis in five of these cohorts ranged from 4.5–21% [28, 50–52]. None of our studies estimated individual or lifetime exposures, however the average RCS exposures (0.19–89.5 mg/m³) and silicosis prevalence (11–37%) are higher than the LSM cohorts. It is of concern that silicosis prevalence was so high, despite the short exposure durations and latency in our review. This observation may reflect an increased risk of silicosis at higher intensities of RCS, as has been observed in miners [28, 53] and other silica exposed industries [54, 55].

A recent review of 12 national surveys in Africa found TB prevalence ranged from 0.11–0.63% [56]. Longitudinal studies of gold miners in large-scale South African gold mines reported annual incidence estimates of between 600 per 100,000 in the very early HIV era to over 2000 per 100,000 during the late 1990s [57–59]. In our review, although estimates varied, TB prevalence among ASM is likely higher than the general population and possibly comparable to gold miners in South Africa during the peak of the HIV pandemic, prior to ART introduction. One included study reported a higher TB prevalence among the peri-mining community than miners (7.9% vs 6.1%) [36] and community prevalence from other studies,

**Table 3. Risk of bias assessed by the Joanna Briggs Institute checklist of studies reporting prevalence data.**

| Study | Sample frame appropriate | Recruitment process | Sample size | Subject and setting description | Sufficient coverage | Valid outcome methods | Standardised outcome measurement | Appropriate statistics | Adequate response rate |
|---|---|---|---|---|---|---|---|---|---|
| **Silicosis** | | | | | | | | | |
| Tse 2007 [31] | Yes | Yes | Yes | Yes | Yes | Yes | Yes | Yes | Yes |
| Souza 2017 [32] | Yes | No | Yes | Yes | Unclear | Yes | Yes | Yes | No |
| Souza 2021 [33] | Yes | Unclear | Yes | Yes | Unclear | No | Unclear | Unclear | Unclear |
| Moyo 2021 [34] | Yes | Unclear | Yes | Yes | Unclear | Unclear | Yes | Yes | Unclear |
| Moyo 2022 [35] | Yes | No | Yes | Yes | Unclear | Unclear | Yes | Yes | Unclear |
| Mbuya 2023 [36] | Yes | Yes | Yes | Yes | Yes | No | Yes | Yes | Unclear |
| **TB** | | | | | | | | | |
| Rambiki 2020 [37] | Yes | No | Yes | Yes | Unclear | Yes | Unclear* | Yes | Unclear |
| Moyo 2021 [34] | Yes | Unclear | Yes | Yes | Unclear | Yes | Unclear* | Yes | Unclear |
| Ohene 2021 [38] | Yes | No | Yes | No | Unclear | Yes | Unclear* | Yes | Unclear |
| Moyo 2022 [35] | Yes | No | Yes | Yes | No | Yes | Unclear* | Yes | Unclear |
| Abeid 2022 [39] | Yes | No | Yes | No | Unclear | Yes | Unclear* | Yes | Unclear |
| Mbuya 2023 [36] | Yes | Yes | Yes | Yes | Yes | Yes | Unclear* | Yes | Unclear |
| **Dust** | | | | | | | | | |
| Osim 1999 [40] | Yes | No | No | Yes | Unclear | Yes | Yes | Yes | Unclear |
| Bratveit 2003 [41] | Yes | No | No | Yes | No | Yes | Yes | Yes | Unclear |
| Tse 2007 [31] | Unclear | Unclear | Unclear | Unclear | Unclear | Unclear | Unclear | Yes | Unclear |
| Gottesfeld 2015 [42] | Yes | No | No | Yes | No | Yes | Yes | Yes | Unclear |
| Mayala 2016 [43] | Unclear | No | No | Yes | No | Unclear | Unclear | No | Unclear |
| Leon-Kamamba 2018 [44] | Yes | Yes | No | Yes | No | Yes | Yes | Yes | Unclear |
| Gottesfeld 2019 [45] | Yes | Unclear | No | Yes | Unclear | Yes | Yes | Yes | Unclear |
| Souza 2021 [33] | Yes | Unclear | No | Yes | Unclear | Yes | Yes | No | Unclear |
| **Spirometry** | | | | | | | | | |
| Osim 1999 [40] | Yes | No | No | Yes | Unclear | Yes | Yes | Yes | Unclear |
| Rajaee 2017 [47] | Yes | Yes | No | Yes | Unclear | Yes | Yes | Yes | Unclear |
| Kyaw 2020 [48] | No | Unclear | No | Yes | Unclear | No | Unclear | No | Unclear |
| **Symptoms** | | | | | | | | | |

(Continued)

**Table 3.** (Continued)

| Study | Sample frame appropriate | Recruitment process | Sample size | Subject and setting description | Sufficient coverage | Valid outcome methods | Standardised outcome measurement | Appropriate statistics | Adequate response rate |
|---|---|---|---|---|---|---|---|---|---|
| Leon Kabamba 2018 [44] | Yes | Yes | No | Yes | No | Yes | Yes | Yes | Unclear |
| Ralph 2018 [46] | Yes | No | No | Yes | Unclear | Yes | Yes | No | Unclear |
| Souza 2021 [33] | Yes | Unclear | Yes | Yes | Unclear | Yes | Yes | No | Unclear |
| Abeid 2022 [39] | Yes | No | Yes | No | Unclear | Yes | Yes | Yes | Unclear |

Green shading indicates the study did achieve the quality assessment criteria and therefore is not at risk of bias. Red shading indicates it did not achieve the quality criteria and is therefore at risk of bias. Orange shading indicates it was unclear and therefore the possibility of risk exists.

* In these studies, how many participants provided sputum samples when indicated and/or had Xpert or smear testing is not clear

although lower than ASM [37, 38], was still higher than expected. This points to the public health concern of community transmission from miners [60].

Both silicosis and TB prevalence in our studies may be overestimated. Selection bias clearly resulted in over-representation of TB and silicosis at healthcare facility screening [35]; only one study used a systematic or randomised approach to sampling [36]. Both silicosis and TB may be over diagnosed in the presence of each other on Chest Xray (CXR) particularly, but not exclusively, by less experienced readers [61]. Clinical diagnosis of TB in the presence of symptomatic silicosis is also challenging. The use of Xpert Ultra may be associated with reduced specificity of microbiological TB, particularly in recurrent disease [62]. Conversely, TB and silicosis may be underestimated. Autopsy evidence suggests radiographic silicosis remains systematically underdiagnosed, more so at high RCS exposures [63]. Symptom screening may miss up to 50% of prevalent infectious cases of TB [64]. Furthermore, in most studies, it was not clear how many participants who screened positive were able to provide sputum samples. Finally, healthy worker bias may contribute to underestimation of both silicosis and tuberculosis [32].

Very high heterogeneity in our meta-analysis precludes inference from these results, however consistent finding of high exposures, disease prevalence and symptoms increases confidence in our main finding of a high burden of occupational respiratory disease. Spirometry findings were equivocal, however this does not provide strong evidence against our main finding for three reasons. First, annual spirometry loss among miners may be relatively modest (FEV1 loss 4.3–11 mls/year), highlighting the need for studies with extended follow up [65]. Second, smoking and tuberculosis were significant unadjusted confounders [65], and third, the studies included were small and had methodological issues.

Our models, using defaults informed by good quality observational research, suggest the reported ASM silicosis and TB outcomes are plausible. For example, in our review, an ASM population with a mean cumulative RCS of 8 mg/m$^3$-year is feasible. Using default model values, this results in a silicosis prevalence of 17.4% and an annual TB incidence of 500 per 100,000 persons. That TB incidence appears to be lower than reported in our studies suggests either under-estimated model parameters or unaccounted factors, for example smoking. More importantly, our modelling demonstrates that reducing RCS results in reductions in silicosis and tuberculosis, with greater effects at higher mean exposures. Reductions of 80% in silica

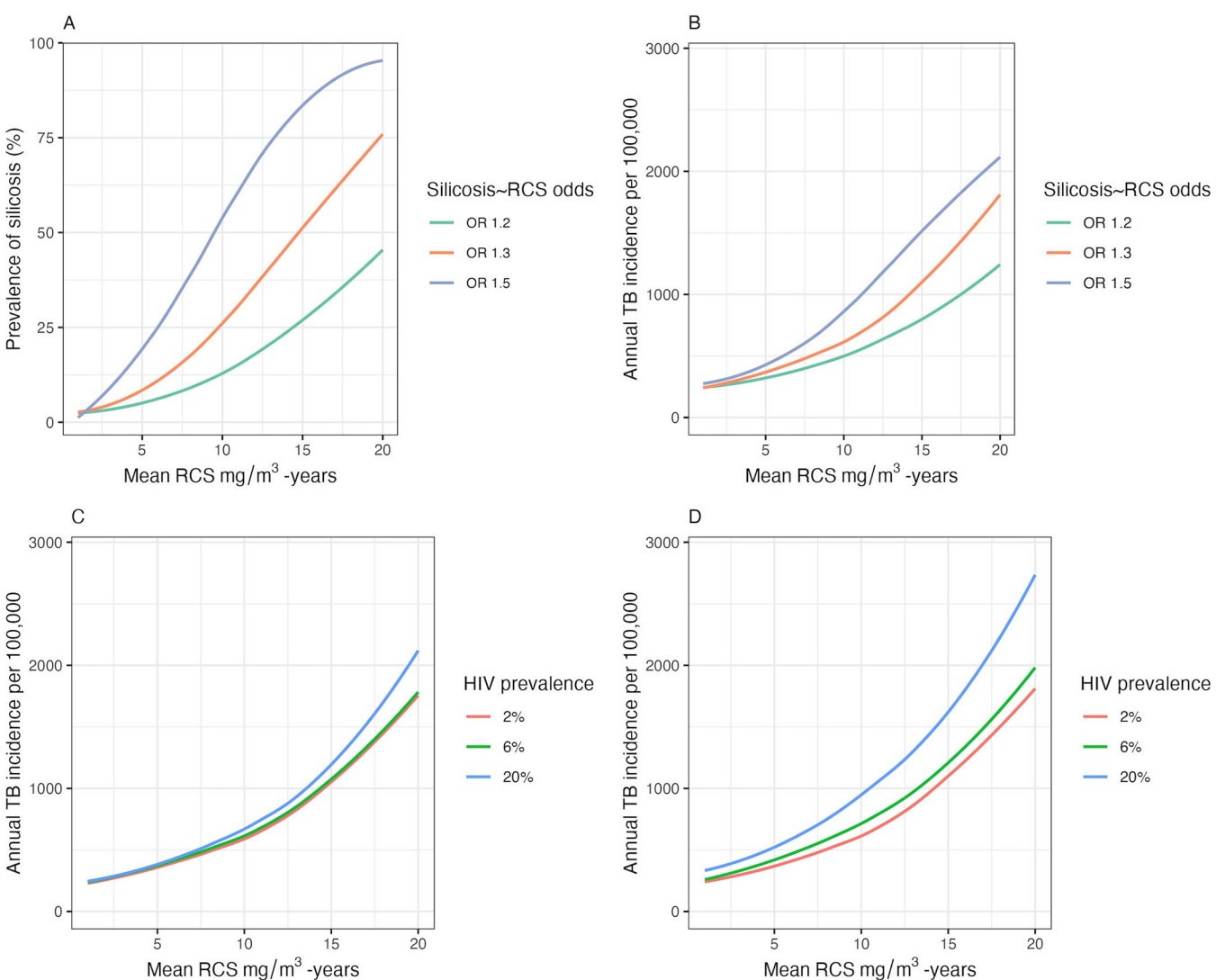

**Fig 5. Simulated estimates of number of silicosis cases and annual tuberculosis (TB) cases at cumulative RCS distributions of increasing mean values.**
Silicosis prevalence and TB incidence was estimated for sample sizes of 10,000 at mean RCS values between 0 and 20, at intervals of 0.01 mg/m$^3$-year. Variables in the model are held constant, unless otherwise stated, at values of: a baseline silicosis prevalence of 2%, an increased odds of 1.3 of silicosis per 1 mg/m$^3$-year increase in RCS exposure, a baseline TB incidence of 200 cases per 100,000 per year, an increased odds of 1.05 of TB per 1 mg/m$^3$-year increase in RCS exposure, an increased odds of TB of 4x in those with silicosis, a randomly distributed baseline prevalence of HIV of 2% and an increased odds of TB of 4x in those with HIV. Curves are smoothed using local polynomial regression, for ease of viewing. In plot A, the silicosis prevalence (%) by distributions of increasing mean cumulative RCS (mg/m$^3$-year) is estimated at three different strengths of association between RCS exposure and silicosis of OR 1.2, 1.3 and 1.5. In plot B, annual tuberculosis cases are estimated at the same three different strengths of association between mean cumulative RCS (mg/m$^3$-year) and silicosis (OR 1.2, 1.3 and 1.5). In plot C, estimates of annual tuberculosis cases are estimated at distributions of increasing mean cumulative RCS (mg/m$^3$-year), at HIV prevalence of 2%, 6% and 20%, the modelled association between HIV and TB is an OR of 2. In plot D, annual tuberculosis cases are again estimated at distributions of increasing mean cumulative RCS (mg/m$^3$-year) and an HIV prevalence of 2%, 6% and 20%, however the strength of association between HIV and TB is an OR of 4.

intensity among ASM and other informal silica-exposed industries are achievable [45, 66]. Furthermore, the introduction of adequate ventilation was estimated at $4000 per ASM mine [43]. Thus, relatively low-cost reductions leading to lower cumulative exposures are possible and, when viewed in the context of our modelling, may result in meaningful reductions in silicosis and TB. If combined with intermittent TB preventative therapy, which leads to temporary reductions in TB incidence [67], and CXR screening, this may lead to sustained and

significant reductions in TB incidence. There exists consensus among experts that TB in mining populations requires a multisectoral approach [6]. The challenges of doing this however should not be underestimated; ASM populations are often hypermobile and operate in hard-to-reach and historically underserved areas.

A single author performed data extraction which was checked by a second author. Given the relatively straightforward outcomes of this review we believe the method is robust. Full text agreement between PH and BS was fair; this was due to different interpretation regarding inclusion a composite measure of respiratory symptoms which was subsequently resolved. We only found one non-English article, which we were able to translate, suggesting that our search strategy may have a language bias. Whilst our search strategy included specific terms it is possible that, by including 5 outcomes, specificity may have been compromised, however only 1 additional study was found in references. Reflecting ambiguity in the definition of ASM, studies may include ASM but not specifically refer to them in their abstract or methods. For example, one study described "small scale" sandstone mines from Rajasthan, India in their introduction but did not meet our search criteria. The study reported a silicosis prevalence (ILO $\geq$1/0) of 275/526 (52%) [68]. A separate study of the sandstone miners from Rajasthan does not make any reference to "small scale" nor informal mining [69]. Definition therefore remains an issue. Importantly however published reports from informal mining and other silica-exposed industries in India, suggest a prevalence at least as high as the studies in our review in most cases [70]. Finally, whilst logistic regression models of RCS and silicosis are valid [24, 51, 71], some studies use alternative modelling distributions which may have yielded different results [50, 72]. For example, the Weibull distribution attenuates risk at higher exposures, although existence of this attenuation for silicosis is contested [73]. It is possible, however, that lower RCS values ($<10$ mg/m$^3$) may underestimate silicosis risk compared to a Weibull model, or vice-versa. We are not aware of studies which adjust for silicosis, RCS and HIV in TB, as in our model, however our conceptual understanding would suggest that combining estimates from studies which model silicosis and RCS, and silicosis and HIV respectively is valid; particularly in high exposure settings [63]. As expected, given the scale of the small-scale mining the number of articles is few and, generally, the study quality limits inference. This highlights the obvious need for future representative and higher quality studies.

## Conclusions

Our review found a uniformly high burden of silicosis and tuberculosis among ASM, despite generally short exposures and latent periods. This burden is likely driven by high RCS exposures which, if reduced, would lead to reductions in silicosis prevalence and tuberculosis incidence. Based on our review, we present five recommendations to policymakers and funders attending the 2$^{nd}$ UN High Level Meeting on TB in September 2023, that build on declarations made in the 2018 meeting [74].

1. RCS control is key and we must enact, fund and enforce RCS exposure limits.

2. Routes to improve ASM worker rights including access to compensation should be promoted.

3. We must fund and support national industrial hygiene programs that have specific remits to include ASM and record and maintain routinely collected data.

4. These programs should work with National TB programs in ASM communities, particularly to scale up TB preventative therapy and diagnostics.

5. We must bring up-to-date centuries old research on the relationship between silicosis and TB, particularly at higher intensity exposures, with the aim of better therapeutics and diagnostics.

Given the scale of the burden, availability and cost of interventions and benefits of a healthy ASM workforce, there can be fewer more effective areas for interventions.

## Supporting information

**S1 File. Search strategy.**
(DOCX)

**S2 File. PRISMA Systematic review and abstract checklist.**
(PDF)

**S3 File. Data extraction and quality assessment template.**
(DOCX)

**S4 File. Detailed description of modelling parameters.**
(DOCX)

**S1 Fig. Line graph of estimated probability of silicosis in increasing categories of 0.5 mg/m3-years at varying associations of cumulative RCS exposure and silicosis.** Incidence is based on values of RCS increasing from 0 to 20 mg/m$^3$-years in increments of 0.1 mg/m$^3$-years in which the odds of silicosis for an increase in 1 mg/m$^3$-years RCS exposure are varied between 1.1 and 2. Note: The true nature of the exposure-outcome relationship will not only depend on the cumulative exposure but also the latency and, potentially, the intensity of exposure. For ease of interpretation, we fitted local polynomial regression lines to all simulated values.
(TIFF)

**S2 Fig. Smoothed density histograms of simulated cumulative RCS exposures of increasing mean values.** Distributions are calculated by taking the square root of the expected mean, drawing a normal distribution with a fixed standard deviation of 0.5 mg/m$^3$-years and performing a quadratic transformation (i.e. squared).
(TIFF)

**S3 Fig.** Funnel plot of A. Silicosis, B. Microbiological tuberculosis and C. Clinical tuberculosis studies, to aid assessment of possible publication bias.
(TIFF)

**S1 Table. Characteristics of studies reporting respiratory symptoms (n = 3) and smoking (n = 8) or substance misuse (n = 2) estimates among ASM.**
(DOCX)

**S2 Table. Characteristics of studies reporting spirometry (n = 3) estimates among ASM.**
(DOCX)

**S3 Table.** Estimates of silicosis prevalence (Table A) and tuberculosis incidence (Table B) at cumulative RCS distributions with a mean of 4, 8, 12 and 16 mg/m3-years based on simulation with a sample size of 10,000. Variables in the model are held constant, unless otherwise stated, at values of: a baseline silicosis prevalence of 2%, an increased odds of 1.3 of silicosis per 1 mg/m3-year increase in RCS exposure, a baseline TB incidence of 200 cases per 100,000 per year, an increased odds of 1.05 of TB per 1 mg/m3-year increase in RCS exposure, an increased odds of TB of 4x in those with silicosis, a randomly distributed baseline prevalence of HIV of

2% and an increased odds of TB of 2x in those with HIV.
(DOCX)

## Acknowledgments

The authors would like to thank individual study authors for their responses to queries regarding individual studies. We would also like to thank Dr André Amaral and Professor Sir Anthony Newman-Taylor for their helpful review and comments of the article, and Professor Maia Lesosky for her assistance regarding our modelling approach.

## Author Contributions

**Conceptualization:** Patrick Howlett, Hader Mousa, Bibie Said.

**Data curation:** Patrick Howlett, Hader Mousa, Bibie Said.

**Formal analysis:** Patrick Howlett, Hader Mousa.

**Investigation:** Patrick Howlett, Bibie Said, Alexander Mbuya.

**Methodology:** Patrick Howlett, Hader Mousa, Bibie Said, Johanna Feary.

**Project administration:** Patrick Howlett, Hader Mousa, Bibie Said.

**Resources:** Patrick Howlett.

**Supervision:** Onn Min Kon, Stellah Mpagama, Johanna Feary.

**Visualization:** Patrick Howlett.

**Writing – original draft:** Patrick Howlett.

**Writing – review & editing:** Patrick Howlett, Hader Mousa, Bibie Said, Alexander Mbuya, Onn Min Kon, Stellah Mpagama, Johanna Feary.

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
