## [Decision Letter · Decision Letter 0]

19 Jun 2023

PGPH-D-23-01031

Silicosis, tuberculosis and silica exposure among artisanal and small-scale miners: A systematic review and modelling paper

Dear Dr. Howlett,

Thank you for submitting your manuscript to PLOS Global Public Health. After careful consideration, we feel that it has merit but does not fully meet PLOS Global Public Health’s publication criteria as it currently stands. Therefore, we invite you to submit a revised version of the manuscript that addresses the points raised during the review process.

We look forward to receiving your revised manuscript.

Kind regards,

Palwasha Yousafzai Khan, MBBCh, MSc, PhD

Academic Editor

Journal Requirements:

Additional Editor Comments (if provided):

Reviewers' comments:

Reviewer's Responses to Questions

**Comments to the Author**

1. Does this manuscript meet PLOS Global Public Health’s publication criteria? Is the manuscript technically sound, and do the data support the conclusions? The manuscript must describe methodologically and ethically rigorous research with conclusions that are appropriately drawn based on the data presented.

Reviewer #1: Yes

Reviewer #2: Yes

2. Has the statistical analysis been performed appropriately and rigorously?

Reviewer #1: Yes

Reviewer #2: Yes

3. Have the authors made all data underlying the findings in their manuscript fully available (please refer to the Data Availability Statement at the start of the manuscript PDF file)?

Reviewer #1: Yes

Reviewer #2: Yes

4. Is the manuscript presented in an intelligible fashion and written in standard English?

Reviewer #1: Yes

Reviewer #2: Yes

5. Review Comments to the Author

Reviewer #1: The study by Howlett et al highlights a major public health topic on silicosis, TB in ASM. The paper is well written and has great merit and is likely to gain much traction during the period leading to the UN-HLM in September.

The authors may highlight that there is need for multisectoral approaches involving Ministries of mines and Health-NTPs and Environmental Health specialties in order to implement the recommendations they suggested. Finally, given that ASM are hypermobile and usually operate in hard to reach and underserved areas, TB services (diagnostic and case holding) need to be tailored to the specific needs of this group.

Reviewer #2: General

1. Comments in abstract should also be addressed in the full text, where applicable.

Title and abstract

1. The authors can remove the colon in the title and keep a continuous line.

2. What was the research question behind this systematic review? As such, I could not pin point anything novel in the research question/ objective. Please describe the novelty which the authors targeted.

3. The relationship between silica dust exposure, silicosis and tuberculosis is very well known (https://bmcpublichealth.biomedcentral.com/articles/10.1186/s12889-021-10711-1). What did the authors add to the existing body of evidence?

4. Define ASM in the abstract itself.

5. The prevalence of silicosis in stone mines of Rajasthan (India) is reported between 38-79% (https://doi.org/10.1016/S2213-2600(15)00052-1). The reported pooled prevalence is an underestimate. In India, small-scale mining is more prevalent and mainly in the unorganized sector.

6. “Our modelling demonstrated decreases in RCS result in reductions in silicosis and tuberculosis, with greater effects at higher mean exposures.” – some numbers here would be more informative. This sentence is an interpretation and not the results.

Introduction

1. Silicosis also affects the TB treatment outcomes (https://www.nature.com/articles/s41598-023-30012-4). Authors might consider adding it.

2. Merge smaller paragraphs.

3. How do ASM and LSM differ? Please also define ASM first.

4. Most of the Indian studies focus on ASM as mining is many a times in the informal sector, illegal, and unregulated. Reference to such Indian studies and context would be helpful.

Materials and methods

1. Well-written

Results

1. “Variables in the model are held constant, unless otherwise stated, at values of: a baseline silicosis prevalence of 2%, an increased odds of 1.3 of silicosis per 1 mg/m3-year increase in RCS exposure, a baseline TB incidence of 200 cases per 100,000 per year, an increased odds of 1.05 of TB per 1 mg/m3-year increase in RCS exposure, an increased odds of TB of 4x in those with silicosis, a randomly distributed baseline prevalence of HIV of 2% and an increased odds of TB of 4x in those with HIV.” – what was the basis of these assumptions?

2. Any results on – at what levels of silica dust does the risk of TB increase?

Discussion

1. Well-written. Collaborative TB-silicosis activities can be discussed (https://www.nature.com/articles/s41598-023-30012-4).

2. Need for maintaining data and addressing underreporting of silicosis should be highlighted.

3. Data from periodic medical examinations would also be a ‘mine’ to analyze, provided they are truly reported data.

Conclusion

1. Collaborative TB-silicosis activities can be proposed to be included in national programs.

6. PLOS authors have the option to publish the peer review history of their article (what does this mean?). If published, this will include your full peer review and any attached files.

**Do you want your identity to be public for this peer review?** For information about this choice, including consent withdrawal, please see our Privacy Policy.

Reviewer #1: No

Reviewer #2: **Yes: **Mihir Rupani

---

## [Decision Letter · Decision Letter 1]

16 Aug 2023

PGPH-D-23-01031R1

Silicosis, tuberculosis and silica exposure among artisanal and small-scale miners: A systematic review and modelling paper

Dear Dr. Howlett,

Thank you for submitting your manuscript to PLOS Global Public Health. After careful consideration, we feel that it has merit but does not fully meet PLOS Global Public Health’s publication criteria as it currently stands. Therefore, we invite you to submit a revised version of the manuscript that addresses the points raised during the review process.

We look forward to receiving your revised manuscript.

Kind regards,

Najmul Haider, PhD

Academic Editor

Journal Requirements:

2. Please insert an Ethics Statement at the beginning of your Methods section, under a subheading 'Ethics Statement'. It must include:

1) The name(s) of the Institutional Review Board(s) or Ethics Committee(s)

2) The approval number(s), or a statement that approval was granted by the named board(s) 

3) (for human participants/donors) - A statement that formal consent was obtained (must state whether verbal/written) OR the reason consent was not obtained (e.g. anonymity). NOTE: If child participants, the statement must declare that formal consent was obtained from the parent/guardian.

3. We noticed that you used "unpublished" in the manuscript. We do not allow these references, as the PLOS data access policy requires that all data be either published with the manuscript or made available in a publicly accessible database. Please amend the supplementary material to include the referenced data or remove the references.

Additional Editor Comments (if provided):

Reviewers' comments:

Reviewer's Responses to Questions

**Comments to the Author**

1. If the authors have adequately addressed your comments raised in a previous round of review and you feel that this manuscript is now acceptable for publication, you may indicate that here to bypass the “Comments to the Author” section, enter your conflict of interest statement in the “Confidential to Editor” section, and submit your "Accept" recommendation.

Reviewer #1: All comments have been addressed

Reviewer #2: All comments have been addressed

2. Does this manuscript meet PLOS Global Public Health’s publication criteria? Is the manuscript technically sound, and do the data support the conclusions? The manuscript must describe methodologically and ethically rigorous research with conclusions that are appropriately drawn based on the data presented.

Reviewer #1: Yes

Reviewer #2: Yes

3. Has the statistical analysis been performed appropriately and rigorously?

Reviewer #1: Yes

Reviewer #2: Yes

4. Have the authors made all data underlying the findings in their manuscript fully available (please refer to the Data Availability Statement at the start of the manuscript PDF file)?

Reviewer #1: Yes

Reviewer #2: Yes

5. Is the manuscript presented in an intelligible fashion and written in standard English?

Reviewer #1: Yes

Reviewer #2: Yes

6. Review Comments to the Author

Reviewer #1: No further comments. The authors have addressed my comments.

Reviewer #2: Thank you for making the changes and congratulations on writing a very important paper. Last few words:

1. You might want to correct the spelling on L483 from “challenging” to “challenges”.

2. A recent review has been published which highlights prevalence of silicosis and silico-tuberculosis in India (https://occup-med.biomedcentral.com/articles/10.1186/s12995-023-00379-1). The authors might want to go through it and the references which it cites for inclusion in the main text or analysis, if felt important. The high prevalence of silicosis and TB in the mining population as compared to other occupations can be considered to be highlighted.

7. PLOS authors have the option to publish the peer review history of their article (what does this mean?). If published, this will include your full peer review and any attached files.

**Do you want your identity to be public for this peer review?** For information about this choice, including consent withdrawal, please see our Privacy Policy.

Reviewer #1: No

Reviewer #2: **Yes: **Mihir Rupani

---

## [Editor Report · Decision Letter 2]

1 Sep 2023

Silicosis, tuberculosis and silica exposure among artisanal and small-scale miners: A systematic review and modelling paper

PGPH-D-23-01031R2

Dear Dr Howlett,

We are pleased to inform you that your manuscript 'Silicosis, tuberculosis and silica exposure among artisanal and small-scale miners: A systematic review and modelling paper' has been provisionally accepted for publication in PLOS Global Public Health.

Best regards,

Najmul Haider, PhD

Academic Editor